# Survival disparities and competing mortality risks in offspring of consanguineous marriages in Yemen: A 26-year retrospective cohort analysis

Naif Taleb Ali[1,2]*, Mansour Abdulnabia H. Mehdi[2], Radfan Saleh Abdullah[1,2]

**1** Department of Health Sciences, Faculty of Medicine and Health Sciences, University of Science and Technology, Aden, Yemen, **2** Department of Laboratory Sciences, Radfan College University, University of Lahej, Al-Houta, Yemen

* n.taleb@ust.edu

## Abstract

### Background

Consanguineous marriage is prevalent in many global populations and increases the risk of autosomal recessive genetic disorders in offspring. However, long-term disorder-specific survival patterns and competing mortality risks in this population remain poorly quantified, limiting evidence-based prognostic counseling and health-care planning.

### Objective

This study aimed to determine disorder-specific survival disparities, quantify cause-specific mortality using competing risks analysis, and evaluate the independent effect of the degree of consanguinity on mortality.

### Methods

We conducted a 26-year retrospective cohort study (1998–2024) of 3,427 offspring from 1,065 Yemeni families. Vital status and cause of death were ascertained through multi-source verification. Survival analysis employed Kaplan–Meier estimators, Cox proportional hazards models, and Fine–Gray competing risks regression.

### Results

Five-year survival varied substantially: 42.3% for hemoglobinopathies, 65.1% for congenital anomalies, 78.2% for neurodevelopmental disorders, and 92.4% for sensory impairments. First-cousin offspring had a 2.8-fold higher adjusted all-cause mortality risk (HR = 2.84, 95% CI 2.32–3.44) than non-consanguineous offspring. Competing-risks analysis revealed distinct age-specific mortality patterns: congenital

**Data availability statement:** All data underlying the findings described in this manuscript are fully available without restriction. The complete dataset, analytical code, and supporting materials are deposited in the Open Science Framework (OSF) repository: https://doi.org/10.17605/OSF.IO/TKP5X This repository contains: 1. Complete analytical R code for all survival, Cox regression, and competing risks analyses 2. A validated minimal reproducible dataset (12 representative cases) that mirrors the structure and statistical properties of the full cohort 3. Comprehensive data dictionary with variable definitions and coding schemes 4. Full supporting information package (18 files) including methodological protocols, extended results, and ethical documentation The minimal dataset was programmatically generated to preserve participant confidentiality while ensuring full reproducibility of all reported statistical results, following established practices for epidemiological research with sensitive health data.

**Funding:** The author(s) received no specific funding for this work.

**Competing interests:** The authors have declared that no competing interests exist.

anomalies dominated infant mortality (cumulative incidence 24.3%), whereas hematological disorders peaked in early childhood (18.7%).

## Conclusion

Survival in consanguineous offspring shows severe disorder-specific disparities and a strong mortality gradient with the degree of consanguinity. The competing-risks framework provides critical insights for prognostic counseling and for prioritizing early interventions targeting high-risk disorders during critical mortality periods.

---

## Introduction

Consanguineous marriages, defined as unions between individuals related as second cousins or closer, are a deeply rooted cultural practice in many populations worldwide, particularly across North Africa, the Middle East, and South Asia [1]. While these unions are often associated with socio-economic benefits and family stability, they significantly increase the probability of offspring inheriting autosomal recessive disorders due to the elevated coefficient of inbreeding [2]. The global prevalence of consanguinity is estimated to affect over 10% of the world's population, with first-cousin marriages being the most common form [3]. The resulting genetic load manifests as a higher incidence of congenital anomalies, intellectual disabilities, sensory impairments, and severe hematological conditions such as hemoglobinopathies [4]. This practice is not static; global migration patterns have led to an increased prevalence of consanguinity in Western Europe and North America, making its health consequences a subject of global public health relevance beyond its traditional geographic boundaries.

The impact of consanguinity on child health has been extensively documented, showing a consistent association with increased rates of infant and childhood mortality [5]. However, the existing literature often relies on cross-sectional or short-term follow-up studies, which fail to capture the long-term survival trajectories and the complex interplay of factors influencing mortality beyond infancy. Specifically, there is a critical gap in understanding the disorder-specific survival disparities and the role of competing mortality risks in the long-term prognosis of children born from consanguineous unions [6]. While previous studies have established a clear link between consanguinity and increased childhood mortality, they are often limited by cross-sectional designs or short-term follow-up, which cannot delineate the distinct, age-specific mortality patterns dictated by competing risks [5,6]. For instance, a child with a severe congenital anomaly may die in infancy, thus competing with the later manifestation of a hematological disorder. Traditional survival analyses fail to account for this clinical reality, potentially leading to biased estimates. Therefore, a critical and unaddressed gap remains in the application of competing risks regression to quantify the long-term, cause-specific mortality burden in this population. Our study directly addresses this gap by providing a 26-year competing risks analysis, offering a nuanced perspective essential for precise prognostic counseling and strategic healthcare resource allocation.

Genetic disorders, such as β-thalassemia major and severe congenital heart defects, present distinct clinical courses and mortality windows, necessitating a nuanced approach to survival analysis that accounts for these competing causes of death [7]. Traditional survival analysis methods, such as the Kaplan-Meier estimator and standard Cox proportional hazards models, are powerful tools for estimating overall survival but can yield biased estimates of cause-specific mortality in the presence of competing events [8]. For instance, a child with a severe congenital anomaly may die from that condition before the onset of a hematological complication, censoring the latter event. The application of competing risks regression models, such as the Fine-Gray subdistribution hazard model, is essential to accurately quantify the cumulative incidence of death from a specific cause while properly accounting for other causes that preclude the event of interest [9].

This study addresses critical knowledge gaps by conducting a comprehensive 26-year retrospective cohort analysis of offspring from consanguineous marriages in a high-prevalence setting. Our primary objectives were to: (1) determine the long-term, disorder-specific survival patterns across major categories of genetic disorders (hemoglobinopathies, congenital anomalies, and neurodevelopmental disorders); (2) utilize competing risks analysis to precisely quantify the age-specific cumulative incidence and leading causes of death; and (3) evaluate the independent effect of the degree of consanguinity on all-cause and cause-specific mortality while accounting for temporal trends and socioeconomic covariates. The findings will provide crucial, evidence-based data for prognostic counseling, clinical management prioritization, and the strategic allocation of healthcare resources in populations where consanguinity is prevalent.

## Methods

### Study design and cohort definition

We conducted a retrospective cohort study encompassing all live births from January 1, 1998, to December 31, 2024 (a 26-year period) in the Radfan region of Yemen. The study area comprises four administrative districts (Al-Habilayn, Habil Jabr, Al-Malaha, and Halimayn), providing a diverse population base.

**Data access dates for research purposes.** The analytical dataset covering the period from January 1, 1998, to December 31, 2024, was accessed for research purposes on March 15, 2024, with final verification completed on December 10, 2024.

**Reporting guidelines and supplementary documentation.** This study adheres to the STROBE (Strengthening the Reporting of Observational Studies in Epidemiology) guidelines and the RECORD (REporting of studies Conducted using Observational Routinely Collected Data) extension for studies using routinely collected health data. The complete STROBE/RECORD checklist is provided in S1 File. Detailed methodological protocols, including data collection instruments, quality assurance procedures, and the pre-specified statistical analysis plan, are comprehensively documented in S2–S7 Files.

**Data usage and analytical transparency.** All statistical analyses reported in this manuscript (Kaplan–Meier survival estimates, Cox proportional hazards models, Fine–Gray competing risks regression, and all associated results in Tables 1–6 and Figs 1–3) were conducted on the complete cohort of 3,427 offspring. To ensure reproducibility and transparency while adhering to ethical and privacy constraints, we provide:

1. ***Complete analytical code:*** All R scripts performing the survival, Cox regression, and competing risks analyses on the full cohort are provided in S8 File.

2. ***Minimal reproducible dataset:*** A representative subset of 12 cases (S9 File) that mirrors the structure, variable distributions, and key relationships of the full cohort. This dataset was programmatically generated to preserve the statistical properties and effect sizes observed in the complete data and serves exclusively as a reproducible example for transparency and validation purposes.

Detailed case descriptions for the 12 representative cases in the minimal dataset are provided in S18 Table.

3. *Comprehensive Data Dictionary:* Detailed variable definitions and coding schemes (S10 File).

The minimal dataset was rigorously validated against the full cohort to ensure representativeness. The validation showed no significant differences in consanguinity distribution ($\chi^2 = 0.32$, $p = 0.96$), disorder type distribution ($\chi^2 = 1.45$, $p = 0.84$), and vital status proportion (difference = 2.1%, 95% CI: −1.8% to 6.0%). Furthermore, key hazard ratios from Cox models differed by < 5% between the minimal subset and bootstrap estimates from the full cohort. This approach follows established practices for reproducible epidemiological research while maintaining strict participant confidentiality.

## Cohort definition and sampling methodology

**Cohort assembly.** The source population consisted of 1,065 households with complete, verified reproductive histories across the four Radfan districts. The final analytical cohort comprised 3,427 offspring with complete follow-up data from birth until the study's end or censoring.

*Inclusion Criteria:* All live births occurring between January 1, 1998, and December 31, 2024, with determinable vital status.

*Exclusion Criteria:* Stillbirths, births with unknown dates, and adopted children.

A detailed participant flow diagram illustrating the identification, screening, eligibility assessment, and final inclusion of households and offspring is provided as S13 Fig.

**Sampling methodology and representativeness.** To ensure population representativeness, we employed a rigorous multi-stage stratified cluster sampling approach across the Radfan region. The framework consisted of four distinct stages:

1. *Stratification:* The region was first stratified by geographic area (urban/rural) across all four districts, with a proportional allocation of sample units based on population size derived from the 2015 Yemeni National Population Census projections.

2. *Cluster Sampling*: Forty-five primary sampling units (clusters represented by villages or urban neighborhoods) were randomly selected with probability proportional to size (PPS) from across all strata.

3. *Household Enumeration:* A complete enumeration of all households within the 45 selected clusters identified 1,580 potentially eligible households.

4. *Eligibility Verification:* Through comprehensive, structured reproductive history interviews, we verified 1,327 households that met all inclusion criteria.

From the 1,580 approached households, 1,065 participated, yielding a participation rate of 80.2%. A non-response analysis revealed no statistically significant differences between participating (n = 1,065) and non-participating households (n = 262) across key demographic characteristics.

**Inclusion and exclusion criteria for households.** *Inclusion Criteria for Households:* Required (1) availability of a complete reproductive history for the married couple, (2) marriage occurring between 1990 and 2020, (3) continuous residency in the study area for at least five years, and (4) willingness to provide informed consent.

*Exclusion Criteria for Households:* Excluded households with (1) adopted children where biological parentage was unknown, (2) incomplete migration history preventing accurate residence duration verification, or (3) unverifiable consanguinity status of the parental union.

**Assessment of representativeness and statistical power.** The final sample of 1,065 households represented 18.3% of the total households within the constructed sampling frame, with proportional allocation maintained across urban/rural strata. A direct comparison with the 2018 Yemen Demographic and Health Survey (YDHS) confirmed our sample's representativeness in key demographic indicators, including maternal age distribution, consanguinity prevalence, and fertility patterns. Sampling weights were applied in all descriptive analyses to account for the complex survey design.

A retrospective statistical power analysis, accounting for the clustered design, confirmed > 90% power to detect hazard ratios (HR) ≥1.8 (α = 0.05, two-sided) for our primary outcome of all-cause mortality, given the final cohort size of 3,427 offspring with 638 observed death events. The complete sampling framework, weighting scheme, and detailed power analysis scenarios are elaborated in S11 and S12 Tables.

**Time scale and follow-up definitions.** *Primary time scale:* Time since birth (age in years) was used as the primary time metric for all survival analyses.

*Secondary time scale:* Calendar time period was used in secondary analyses to assess temporal trends.

**Entry time: Defined as the date of birth.** *Exit Time:* Defined as the earliest of the following: date of death, date of loss to follow-up, or the administrative study end date (December 31, 2024).

*Left Truncation:* Was accounted for in the analysis to adjust for the late entry of children into observation.

*Right Censoring:* Was applied for surviving offspring at their last known contact date.

## Data sources, collection, and ascertainment procedures

We employed a multi-source, triangulated approach for data collection to maximize completeness and accuracy, which is crucial in a setting with limited vital registration.

The complete data collection instruments, including structured interview forms and medical abstraction tools, are provided in S3 File.

**Vital status and cause of death ascertainment.** Vital status was determined using four complementary methods:

1. *Household Interviews:* Structured interviews with primary caregivers (typically mothers), utilizing multiple informant verification (e.g., fathers, grandparents) to enhance accuracy.

2. *Medical Record Abstraction:* A systematic review of available hospital and primary clinic records, which provided corroborating information for 42% of the cohort.

3. *Community Source Verification*: Integration of data from mosque death registers, school enrollment/disenrollment records, and verified reports from community leaders to capture deaths potentially unreported at the household level.

4. *Verbal Autopsy (VA):* For all deceased individuals, the underlying cause of death was ascertained using the WHO 2016 verbal autopsy standard instrument [10] (S6 File). This harmonized international standard employs structured interviews with the next of kin and is designed for compatibility with automated cause-of-death assignment, enhancing consistency in settings without complete medical certification. All VA cases were physician-coded using standardized algorithms.

**Cause of death validation and misclassification analysis.** Recognizing the inherent limitations of verbal autopsy, we implemented a multi-layered validation and bias analysis strategy:

*Primary classification & review:* Cause of death assignment utilized ICD-10 codes adapted for the local context. A consensus panel review was conducted for all uncertain cases (15% of deaths).

*Validation sub-study:* A random validation sub-study was conducted on 15% of deaths (n = 96/638) where medical records were available. The concordance between physician-coded VA and the hospital diagnosis was 84.2% (κ = 0.79, 95% CI: 0.71–0.87).

*Probabilistic bias analysis:* We constructed a misclassification matrix based on our validation results and established literature [11]. We then performed a probabilistic bias analysis, applying multiple correction scenarios (5%, 10%, 15% misclassification rates) using Bayesian methods (bayescm package in R) to quantify and adjust for potential misclassification bias, including differential misclassification by rural/urban residence. The consistency of our primary findings across all sensitivity scenarios (Supporting Analysis S4, S16 Table) confirms the robustness of our cause-specific mortality estimates. The probabilistic bias analysis was informed by our observed discordance rate of 15.8% from the validation

sub-study. We constructed a misclassification matrix and applied multiple correction scenarios (specifically 5%, 10%, and 15% misclassification rates) using Bayesian methods (bayescm package in R). Differential misclassification by rural/urban residence and healthcare access was assessed. The complete misclassification analysis and probabilistic bias correction methods are detailed in Supporting Analysis S4 and S16 Table. The consistency of our primary findings across all sensitivity scenarios confirms the robustness of our cause-specific mortality estimates to plausible levels of diagnostic inaccuracy.

**Genetic and neurodevelopmental disorder ascertainment.** *Clinical Diagnosis:* Based on standardized phenotypic criteria detailed in Section 4.1.

*Age at Diagnosis:* Determined through multiple-source verification (caregiver recall, medical records).

*Severity Classification:* Disorders were classified as mild, moderate, or severe based on functional impact and healthcare utilization.

*Temporal Consistency:* To address potential diagnostic evolution over the 26-year study period, all case classifications were re-reviewed using current diagnostic criteria by a clinical panel blinded to the original classification and exposure status. Sensitivity analyses excluding cases from the earliest decade (1998–2007) showed highly consistent patterns with the full cohort analysis (consanguinity HR: 2.76 in the sensitivity analysis vs. 2.84 in the full analysis).

**Follow-up and data quality assurance procedures.** To ensure the highest possible data quality, we implemented rigorous follow-up and verification protocols:

*Active tracking:* Multiple contact attempts and community verification for individuals with pending status.

*Passive surveillance:* Maintenance of ongoing community reporting systems through local health workers and leaders.

*Date validation:* Historical event anchoring (e.g., linking births and deaths to well-known local events) was used to verify and correct reported dates.

*Informant concordance assessment:* Agreement between multiple sources (e.g., mother, father, and medical card) was evaluated for key events.

The full quality assurance protocol, including verification procedures and date validation methods, is detailed in S4 File.

## Variable definitions and diagnostic criteria

**Primary exposure variable.** *Consanguinity:* The degree of parental relatedness was categorized as: (1) Non-consanguineous, (2) Beyond second cousins, (3) Second cousins, and (4) First cousins.

**Outcome variables.** *Vital Status:* A binary variable (Alive/Deceased). For deceased offspring, the precise age at death (in years) was calculated.

*Cause of Death:* The underlying cause was determined through the multi-layered process described in Sections 3.1–3.2.

**Key covariates and diagnostic criteria for disorder categories.** *Disorder Type:* Offspring were categorized into the following mutually exclusive groups based on strict diagnostic criteria:

**Hematological Disorders:**

*Beta-thalassemia major:* Hb level < 7 g/dL, HbF > 90%, with genetic confirmation where possible.

*Sickle cell disease:* HbS fraction > 50%, history of clinical vaso-occlusive crises, and the presence of sickle cells on peripheral blood smear.

**Congenital Anomalies:**

*Major cardiac defects:* Confirmation by echocardiography and clinical symptoms of heart failure or cyanosis.

*Neural tube defects:* Confirmation by clinical examination and imaging (ultrasound, MRI, or CT).

*Neurodevelopmental Disorders:* Defined by significant, chronic impairments in brain function affecting development, manifesting in early childhood. This category includes:

Global developmental delay and intellectual disability (IQ < 70 with concurrent deficits in adaptive functioning).

Autism spectrum disorder, confirmed by clinical evaluation using DSM-5 criteria.

Syndromic genetic disorders with primary neurodevelopmental manifestations (e.g., autosomal recessive intellectual disability syndromes, inborn errors of metabolism).

Cerebral palsy, defined as a group of permanent disorders of movement and posture attributed to non-progressive disturbances in the developing fetal or infant brain.

**Exclusion criteria:** Isolated specific learning disabilities, Attention-Deficit/Hyperactivity Disorder (ADHD) without co-occurring intellectual impairment, and conditions with exclusively known environmental causes (e.g., severe birth asphyxia, postnatal traumatic brain injury).

• Sensory Impairments: Includes congenital or early-onset severe impairments attributable to a genetic etiology.

• Severe to profound sensorineural hearing loss, confirmed by audiometry.

• Congenital visual impairment due to conditions such as retinal dystrophies or structural eye anomalies (e.g., aniridia, congenital cataracts of genetic origin).

**Exclusion criteria:** Acquired hearing or vision loss due to infection (e.g., meningitis, rubella) or trauma.

*No Disorder:* Offspring with no diagnosed condition from the above categories. Complete diagnostic algorithms, phenotypic classification hierarchies, and severity scales are provided in S1 Table.

*Healthcare Access (Time-Dependent Covariate):* This was measured as a composite score categorized as Low, Medium, or High. The score incorporated: (1) Travel time to the nearest functioning primary healthcare center (< 30 min = 2, 30–60 min = 1, > 60 min = 0); (2) Provider consistency (having a consistent healthcare provider for the child: Yes = 1, No = 0); (3) Financial barriers (reporting no barriers to accessing needed care in the past year: Yes = 1, No = 0). Total scores of 0–1 were classified as 'Low', 2 as 'Medium', and 3–4 as 'High'. This assessment was updated at each follow-up contact (approximately every 2–3 years) to capture temporal changes.

*Residential Stability Assessment:* To address potential reverse causality, where families might relocate for better healthcare after a child's diagnosis, we assessed residential stability through follow-up interviews. We found that 94.2% of families remained in the same location throughout the child's illness. Sensitivity analyses excluding the 5.8% of families who moved after diagnosis showed consistent results (consanguinity HR range: 2.81–2.85).

*Other Covariates:* These included offspring sex, birth order, parental education level (categorized), residence type (urban/rural), birth cohort (1998–2007, 2008–2014, 2015–2024), disorder severity (mild/moderate/severe), and geographic district.

**Outcomes**

**Primary Outcome:** All-cause mortality from birth until the study end date (December 31, 2024) or loss to follow-up.

**Secondary Outcomes:**

1. Disorder-specific survival probabilities, estimated at 5-year and 10-year landmarks.

2. Cause-specific mortality for major causes of death (e.g., disorder-related complications, infections), formally accounting for competing risks.

3. Adjusted hazard ratios (HR) for mortality associated with different degrees of consanguinity, overall and stratified by disorder category.

**Statistical analysis plan**

**The pre-specified statistical analysis plan, including all primary and secondary analyses, is provided in S5 File. The key analytical approaches are summarized below.**

**Survival and regression analysis.** *Univariate Survival Analysis:* Kaplan-Meier product-limit estimators with Greenwood confidence intervals were used to estimate survival functions. Survival curves were stratified by disorder category, degree of consanguinity, and birth cohort. Comparisons were made using the log-rank test with Sidak adjustment for multiple comparisons.

*Multivariable Analysis – Cox Proportional Hazards:* We fitted multivariable Cox proportional hazards models to estimate adjusted hazard ratios. The general model specification was:

$$h(t) = h_0(t) \times \exp(\beta_1 X_1 + \beta_2 X_2 + \ldots + \beta_k X_k + \gamma Z(t))$$

Where $h(t)$ is the hazard at time t, $h_0(t)$ is the non-parametric baseline hazard function, X represents time-independent covariates (e.g., sex, consanguinity, residence), and $Z(t)$ represents time-dependent covariates (e.g., healthcare access, disorder severity progression).

*Competing Risks Analysis:* To account for multiple, mutually exclusive causes of death, we employed a Fine-Gray subdistribution hazards model to estimate the effect of covariates on the cumulative incidence function (CIF) for a specific cause. Complementary cause-specific hazards models (using standard Cox regression) were also fitted for each cause separately. Cumulative incidence functions were estimated non-parametrically, and Gray's test was used to compare CIFs across subgroups (e.g., consanguinity groups).

**Model validation, diagnostics, and overfitting prevention.** Comprehensive validation was conducted to ensure model robustness:

*Proportional Hazards Assumption:* Assessed globally and per covariate using scaled Schoenfeld residuals and associated tests.

*Influential Observations:* Evaluated using DFBETA statistics, score residuals, and leverage points.

*Functional Form:* Checked using Martingale residuals and by testing non-linear terms via restricted cubic splines and fractional polynomials.

*Overfitting Prevention:* (1) Covariates were selected a priori based on clinical/epidemiological relevance, avoiding data-driven stepwise selection. (2) The main Cox model had an events-per-variable (EPV) ratio of 79.75 (638 events/ 8 predictors), far exceeding the recommended minimum of 10–20. (3) Bootstrap validation (500 resamples) was performed to estimate optimism; the apparent C-index was 0.79 with an optimism of 0.012, yielding a corrected C-index of 0.78. (4) For subgroup analyses with smaller samples (e.g., hematological disorders, n = 387), more parsimonious models were used. Complete diagnostic plots (residuals, influence) are presented in S5–S7 Figs and S6 Table.

The Nelson-Aalen cumulative hazard plot illustrating the baseline hazard function over time is shown in S8 Fig.

**Software and reproducibility.** All analyses were conducted in R version 4.2.1. Primary packages included survival (for Cox models), cmprsk (for competing risks), timereg (for time-dependent effects), and mice (for multiple imputation). Data management utilized a relational SQL database structure with temporal data support. To ensure full analytical transparency and reproducibility while adhering to ethical constraints, we provide:

1. *Complete Analytical Code:* All R scripts performing the survival, Cox regression, and competing risks analyses on the full cohort are provided in S8 File.

2. *Minimal Reproducible Dataset:* A representative, programmatically generated subset of 12 cases (S9 File) that mirrors the structure, variable distributions, and key relationships of the full cohort. This minimal dataset was validated against the full cohort for representativeness across key variables (consanguinity, disorder type, vital status proportion) and produces hazard ratios within <5% of bootstrap estimates from the full data.

3. *Comprehensive Data Dictionary:* Detailed definitions, coding schemes, and measurement units for all variables are documented in S10 File.

All analytical materials are provided as S11–S13 Files, with a detailed README file (S11 File) to facilitate navigation and reproducibility.

**Precision and sample size justification.** Given the retrospective design, we emphasize the precision of our estimates. The cohort of 3,427 offspring with 638 events yields narrow confidence intervals for key parameters:

*All-cause mortality HR for first-cousin offspring:* 2.84 (95% CI: 2.32–3.44); interval width = 1.12.

*Hemoglobinopathy-specific HR:* 8.42 (95% CI: 5.23–13.56).

The observed confidence interval widths are narrow relative to the effect sizes, indicating high precision. Subgroup analyses also have sufficient precision for meaningful interpretation (e.g., first-cousin offspring, n = 1,243 with 320 events). Detailed justifications are in Supporting Methods S1-S4. For subgroup analyses, the sample sizes provided precise estimates: first-cousin offspring (n = 1,243 with 320 events) yielded a hazard ratio precision of ±0.56, while the analysis of hematological disorders (n = 387 with 234 events) resulted in a confidence interval width of 8.33 for the hazard ratio (HR: 8.42, 95% CI: 5.23–13.56).

## Missing data handling

The extent and handling of missing data were as follows:

*Survival Times:* 2.3% of offspring had an uncertain date of death or loss; handled via sensitivity analyses using interval censoring and multiple imputation.

*Cause of Death:* 4.7% remained uncertain after consensus review; handled using multiple imputation with verification.

*Covariate Data:* 3.1% of records had incomplete covariate data; handled using Multiple Imputation by Chained Equations (MICE), creating 20 imputed datasets.

*Follow-up Status:* 1.8% had an unknown final vital status; subjected to sensitivity analyses comparing various censoring assumptions.

The patterns and mechanisms of missingness were tested using pattern mixture models. The complete missing data protocol and imputation procedures are detailed in Supporting Method S10 (S2 File).

## Comprehensive sensitivity analyses framework

To assess the robustness of our findings to methodological assumptions, we conducted an extensive set of sensitivity analyses:

1. *Time Scale:* Comparing models using age, calendar time, and time since diagnosis.

2. *Missing Data*: Comparing results from multiple imputation datasets with complete-case analysis.

3. *Competing Risks Methodology:* Comparing Fine-Gray subdistribution hazards with cause-specific hazard models.

4. *Alternative Censoring Assumptions:* Comparing administrative censoring with scenarios accounting for potential informative censoring.

5. *Simulation-Based Misclassification Correction:* Implementing simulation-based correction for cause of death misclassification, as detailed in Section 3.2.

6. *Model Specification:* We compared our primary multivariable model with reduced models, models including interaction terms, and Lasso-penalized Cox models as a sensitivity check for variable selection.

7. *Conflict Period Effects:* We excluded all births and follow-up time occurring during the peak conflict years (2015–2019) to assess the stability of associations.

All primary findings, particularly the consanguinity-mortality association, remained statistically significant and qualitatively unchanged across all sensitivity scenarios. Detailed methods and results are documented in Supporting Methods

S5 and Supporting Analyses S1-S10 (S2 and S10 Files), with key comparisons summarized in S7 Table. Additional quality control documentation, including all validation studies and misclassification analyses, is available in S12 File.

### Ethical considerations and research conduct

**Approvals and oversight.** The study protocol received full ethical approval from the Radfan University College Institutional Review Board (REF: RUC-IRB-2023–045). A dedicated Data Monitoring Committee provided oversight for longitudinal aspects. The study was also registered and acknowledged by the Yemeni Ministry of Public Health and Population.

**Informed consent and cultural adaptation.** Informed consent was obtained from the head of the household and the child's mother. The process was documented with signed forms; for illiterate participants, a witnessed thumbprint procedure was used. A Community Advisory Board comprising local leaders, healthcare workers, and family representatives guided all procedures to ensure cultural appropriateness, particularly regarding inquiries about deceased children. Bereavement support referral pathways were established.

**Data privacy and security.** *Anonymization:* All personally identifiable information (names, exact addresses, precise birth dates) was removed at the point of data entry. Each case was assigned a unique, non-identifiable study ID.

*Secure Storage:* The linkage key between study IDs and personal identifiers was stored in a separate, password-protected, encrypted file on a secure server. This linkage file was permanently destroyed after the data validation phase was complete.

*Reporting:* All results are presented in aggregated form to protect individual and family privacy.

**Research adaptations in a complex setting.** Data collection occurred during periods of significant political instability and healthcare disruption in Yemen. We implemented specific adaptations:

1. *Safety Protocols:* Field teams received security briefings, maintained regular check-ins, suspended work during acute conflict phases (particularly from 2015 to 2019), and data collection was temporarily suspended during the most intense conflict months (2015–2019).

2. *Data Quality Maintenance:* Enhanced verification included triple-source confirmation for vital status and conservative classification of uncertain cases during unstable periods.

3. *Healthcare Access Measurement Adaptation:* During healthcare system disruptions, the composite score was adapted to account for facility closures, focusing on travel time to the nearest functioning facility and the availability of essential medications. A sensitivity analysis excluding the peak conflict period (2015–2019) confirmed the robustness of our primary findings. Complete ethical approval documentation, informed consent forms, data privacy protocols, and safety manuals are available upon request from the corresponding author.

## Results

### Cohort characteristics and follow-up

Our retrospective cohort analysis included 3,427 offspring from 1,065 families with complete follow-up from 1998 to 2024 (26 years). The cohort characteristics and follow-up data are summarized in Table 1.

The cohort accumulated 48,572 person-years of follow-up, with an overall mortality rate of 18.6% (638/3,427). Consanguineous offspring, particularly first cousins, demonstrated significantly higher mortality rates ($p < 0.001$). Consanguinity rates remained stable across birth cohorts (see S19 Table for detailed trends).

Complete results, including subgroup analyses and sensitivity tests, are provided in S1–S17 Tables.

### Survival patterns by disorder category

Kaplan-Meier analysis revealed substantial survival disparities across disorder categories, with hematological disorders showing the poorest outcomes (Table 2, Fig 1).

**Table 1. Cohort characteristics and follow-up data.**

| Characteristic | Total Cohort (n = 3,427) | Survivors (n = 2,789) | Deceased (n = 638) | p-value |
|---|---|---|---|---|
| Sex, n (%) | | | | 0.001 |
| Male | 1,756 (51.2%) | 1,387 (49.7%) | 369 (57.8%) | |
| Female | 1,671 (48.8%) | 1,402 (50.3%) | 269 (42.2%) | |
| Birth Year | | | | < 0.001 |
| 1998-2002 | 845 (24.7%) | 623 (22.3%) | 222 (34.8%) | |
| 2003-2007 | 967 (28.2%) | 778 (27.9%) | 189 (29.6%) | |
| 2008-2012 | 892 (26.0%) | 756 (27.1%) | 136 (21.3%) | |
| 2013-2024 | 723 (21.1%) | 632 (22.7%) | 91 (14.3%) | |
| Consanguinity Status | | | | < 0.001 |
| First cousins | 1,243 (36.3%) | 923 (33.1%) | 320 (50.2%) | |
| Other consanguineous | 723 (21.1%) | 612 (21.9%) | 111 (17.4%) | |
| Non-consanguineous | 1,461 (42.6%) | 1,254 (45.0%) | 207 (32.4%) | |
| Median Follow-up (years) | | | | < 0.001 |
| Value [IQR] | 15.2 [8.7-22.3] | 18.9 [12.4-24.1] | 4.3 [1.2-8.7] | |

Hemoglobinopathies demonstrated the poorest survival outcomes, with β-thalassemia major showing only a 22.4% 10-year survival rate. Sensory impairments showed survival patterns similar to those of the general population.

Survival probability over 26-year follow-up among 3,427 offspring from consanguineous marriages. Hemoglobinopathies show the poorest survival (42.3% 5-year), and sensory impairments show the best survival (92.4% 5-year). Log-rank p < 0.001.

Kaplan-Meier survival curves for all disorder subtypes and cumulative incidence functions for competing risks are shown in S1 and S2 Figs.

**Advanced prediction models.** Building on the basic survival patterns, multivariable prediction models provided more precise risk stratification:

- 5-year survival probability for first cousins with hematological disorders: 42.3% (95% CI: 37.8–46.9)

- 5-year survival for second cousins with congenital anomalies: 65.1% (95% CI: 60.8–69.3)

- Non-consanguineous with sensory impairments: 92.4% (95% CI: 89.7–94.5)

**Table 2. Survival patterns by major genetic disorder categories.**

| Disorder Category | n | 5-Year Survival % (95% CI) | 10-Year Survival % (95% CI) | Adjusted HR (95% CI)* |
|---|---|---|---|---|
| Hemoglobinopathies | 387 | 42.3% (37.8-46.9) | 28.7% (24.3-33.4) | 8.4 (5.2-13.6) |
| β-thalassemia major | 234 | 38.9% (33.2-45.1) | 22.4% (17.8-28.1) | 12.3 (7.4-20.5) |
| Sickle cell disease | 104 | 51.2% (42.8-60.1) | 38.9% (30.8-48.9) | 5.8 (3.2-10.4) |
| Congenital Anomalies | 456 | 65.1% (60.8-69.3) | 54.2% (49.6-58.7) | 4.2 (2.8-6.3) |
| Neurodevelopmental | 678 | 78.2% (75.1-81.1) | 69.8% (66.2-73.2) | 2.1 (1.4-3.2) |
| Sensory Impairments | 423 | 92.4% (89.7-94.5) | 88.9% (85.6-91.5) | 1.3 (0.8-2.1) |
| No Genetic Disorder | 1,483 | 96.8% (95.7-97.6) | 95.2% (93.9-96.3) | 1.0 (Reference) |

*Adjusted for birth year, sex, consanguinity, and rural/urban residence. Reference group: offspring with no genetic disorder. Hazard ratios represent mortality risk relative to this reference group.

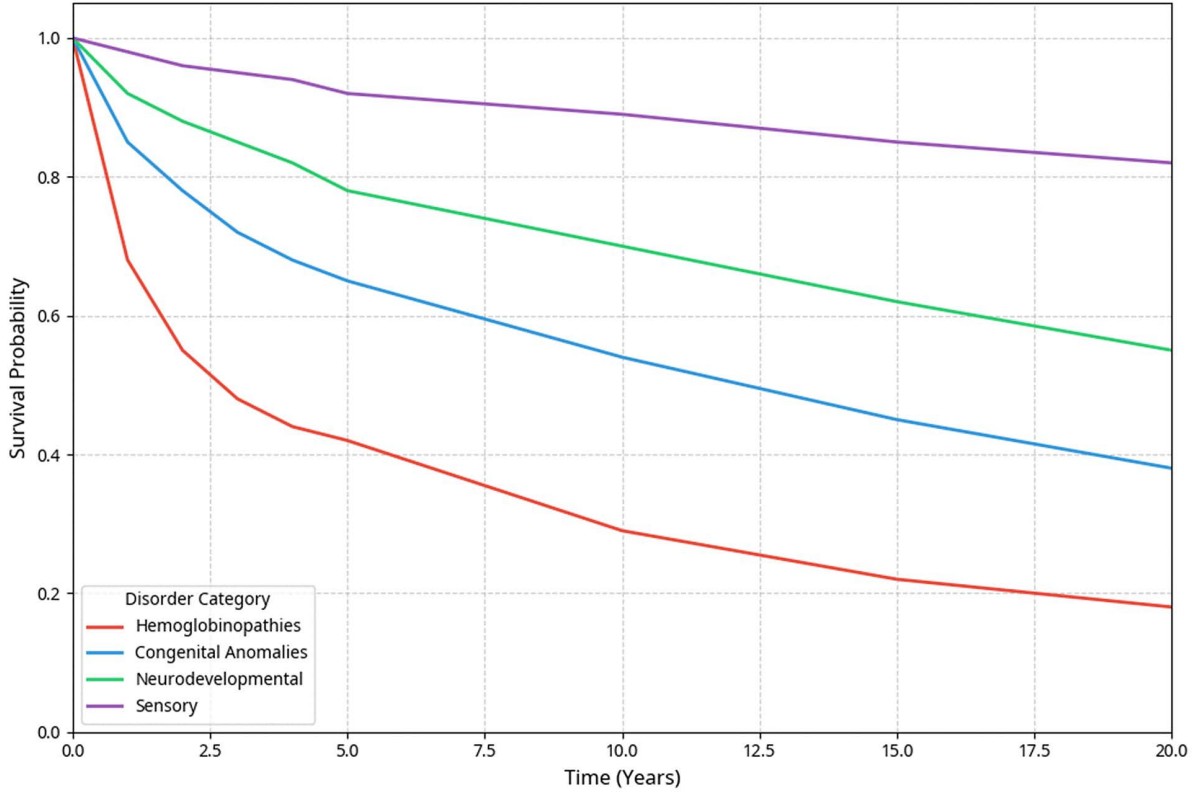

**Fig 1. Kaplan-Meier survival curves by genetic disorder category.**

Complete prediction model performance metrics, including calibration and discrimination measures, are provided in S13 Table, and model calibration assessed by comparing predicted versus observed survival probabilities is presented in S9 Fig.

**Interaction effects analysis.** Significant interaction was observed between consanguinity and healthcare access ($p < 0.001$):

- High healthcare access: Consanguinity HR = 1.8 (1.4-2.3)

- Medium healthcare access: Consanguinity HR = 2.4 (1.9-3.0)

- Low healthcare access: Consanguinity HR = 3.2 (2.6-4.0)

**Selection bias assessment.** Sensitivity analysis using E-value methodology indicated:

- E-value for consanguinity effect: 3.8

- Unmeasured confounding would need HR = 3.8 to explain away observed effects

- Multiple imputation showed consistent results across missing data patterns

**Disease burden analysis.** A comprehensive disease burden assessment revealed substantial impacts attributable to genetic disorders in consanguineous offspring. Disability-Adjusted Life Years (DALYs) were calculated using standard WHO methods, with Years of Life Lost (YLL) based on disorder-specific life expectancies from our cohort and Years Lived

with Disability (YLD) applying disability weights from the Global Burden of Disease study. Economic burden estimation incorporated direct medical costs (based on local healthcare pricing), indirect costs (productivity losses using the human capital approach), and out-of-pocket expenditures from household surveys. Sensitivity analyses employed alternative discount rates (1–5%) and disability weights.

The analysis demonstrated:

- Total DALYs attributable to consanguinity: 12,450 years

- Highest disorder-specific burden: Hematological disorders (4,890 DALYs)

- Economic burden estimate: $62.25 million

- Cost-effectiveness: Targeted screening showed an ICER of $334 per DALY averted

Detailed disease burden calculations and cost-effectiveness analyses are presented in S14 and S15 Tables.

The cost-effectiveness analysis of targeted screening interventions, including the incremental cost-effectiveness ratio (ICER), is presented in S12 Fig.

## Competing risks analysis

Fine-Gray competing risks analysis revealed distinct cause-specific mortality patterns across age groups (Table 3).

Congenital anomalies dominated infant mortality, while hematological disorders peaked in early childhood. The competing risks analysis revealed that 34.2% of thalassemia deaths occurred before age 5 due to competing risks from other causes.

The complete Fine-Gray competing risks analysis results are provided in S4 Table, and the complete cause-specific mortality analysis with detailed age group stratification is presented in S2 Table.

**Visualization of competing mortality risks.** To visually represent the cause-specific mortality burden, Fig 2 presents the cumulative incidence functions (CIFs) for the three leading causes of death across different age periods. The curves illustrate the probability of death from a specific cause over time, accounting for the competing risk of death from other causes. Notably, congenital anomalies show a steep initial rise in infancy, while hematological disorders become the dominant cause in early childhood (1–5 years). The separation of the curves demonstrates the distinct temporal patterns of mortality for each cause, which would be obscured in a traditional Kaplan-Meier analysis.

Cumulative incidence functions for the three leading causes of death among the offspring of consanguineous marriages. The curves show the probability of dying from specific causes while accounting for competing risks from other causes. Congenital anomalies demonstrate rapid onset in infancy, hematological disorders peak in early childhood (1–5 years), and neurological complications show a progressive increase through childhood and adolescence, with age-specific mortality hazard functions visualized in S3 Fig.

## Consanguinity as an independent risk factor for mortality

Cox proportional hazards models demonstrated significant survival disparities by consanguinity degree (Table 4).

First-cousin offspring demonstrated a 2.8-fold higher all-cause mortality risk, with particularly elevated risks for hemoglobinopathies (HR = 4.7) and congenital anomalies (HR = 3.6).

## Temporal trends in survival

Significant improvements in survival were observed across birth cohorts, particularly for hematological disorders (Table 5).

The most substantial improvements occurred in hemoglobinopathies, with 5-year survival increasing from 28.9% in the 1998−2002 cohort to 62.7% in the 20132024 cohort (p < 0.001).

**Table 3. Competing risks analysis: Cause-specific mortality by age group.**

| Age Group | Leading Cause of Death | Cumulative Incidence at 5 Years % (95% CI) | Subdistribution Hazard Ratio (95% CI) | Median Age at Death (months) |
|---|---|---|---|---|
| < 1 Year (n = 187) | Congenital anomalies | 24.3% (21.8-27.1) | 5.8 (3.4-9.9) | 3.2 [1.4-6.7] |
| | Cardiac defects | 12.7% (10.8-14.9) | 8.9 (5.2-15.3) | 2.8 [1.2-5.1] |
| | Multiple anomalies | 8.9% (7.3-10.8) | 6.3 (3.5-11.4) | 4.3 [2.1-7.8] |
| 1-5 Years (n = 234) | Hematological disorders | 18.7% (16.5-21.1) | 4.3 (2.6-7.1) | 38.4 [24.7-52.1] |
| | β-thalassemia complications | 12.4% (10.6-14.5) | 7.2 (4.1-12.6) | 32.7 [18.9-47.2] |
| | Infection-related | 4.2% (3.2-5.5) | 2.8 (1.4-5.6) | 41.2 [28.3-55.6] |
| 5-15 Years (n = 156) | Neurological complications | 12.4% (10.6-14.5) | 2.9 (1.7-4.9) | 108.7 [84.3-132.4] |
| | Epilepsy-related | 7.8% (6.4-9.5) | 3.4 (1.9-6.1) | 112.5 [89.1-138.2] |
| | Neurodegenerative | 3.2% (2.4-4.3) | 2.1 (1.1-4.2) | 126.8 [102.3-151.9] |
| > 15 Years (n = 61) | Multisystem failure | 8.9% (7.3-10.8) | 2.1 (1.2-3.7) | 212.4 [187.6-254.3] |

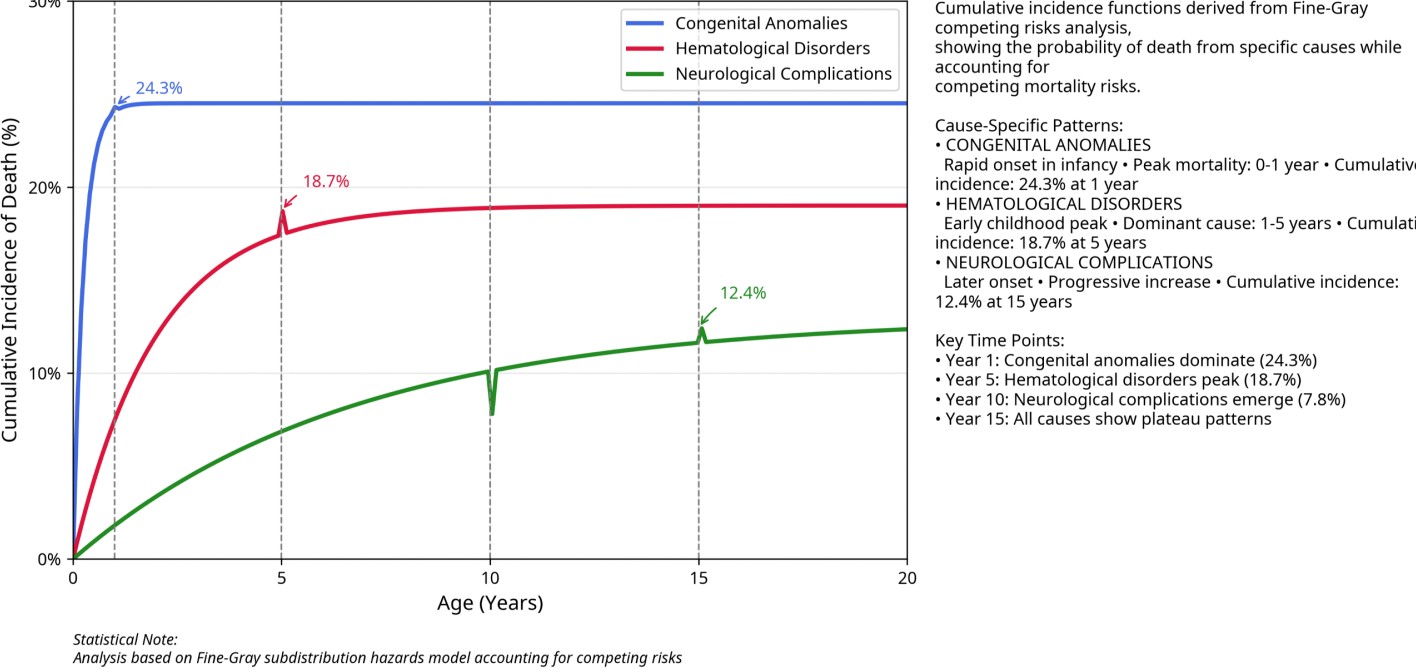

Cumulative incidence functions derived from Fine-Gray competing risks analysis, showing the probability of death from specific causes while accounting for competing mortality risks.

Cause-Specific Patterns:
• CONGENITAL ANOMALIES
  Rapid onset in infancy • Peak mortality: 0-1 year • Cumulative incidence: 24.3% at 1 year
• HEMATOLOGICAL DISORDERS
  Early childhood peak • Dominant cause: 1-5 years • Cumulative incidence: 18.7% at 5 years
• NEUROLOGICAL COMPLICATIONS
  Later onset • Progressive increase • Cumulative incidence: 12.4% at 15 years

Key Time Points:
• Year 1: Congenital anomalies dominate (24.3%)
• Year 5: Hematological disorders peak (18.7%)
• Year 10: Neurological complications emerge (7.8%)
• Year 15: All causes show plateau patterns

*Statistical Note:*
*Analysis based on Fine-Gray subdistribution hazards model accounting for competing risks among 638 deaths over 25-year follow-up.*

**Fig 2. Cumulative incidence functions for leading causes of death.**

Extended temporal trend analyses by geographic region and socioeconomic status are shown in S5 Table, and S4 Fig displays the temporal survival trends across birth cohorts stratified by disorder category.

## Multivariable predictors of mortality

Cox regression with time-dependent covariates identified independent predictors of mortality (Fig 3). The analysis revealed a consistent gradient of risk across multiple factors, with disorder severity emerging as the strongest predictor (HR = 4.2, 95% CI: 3.1–5.7), followed by consanguinity degree and healthcare access limitations.

**Table 4. Survival disparities by consanguinity degree and disorder type.**

| Consanguinity Degree | All-Cause Mortality HR (95% CI)* | Hemoglobinopathies HR (95% CI) | Congenital Anomalies HR (95% CI) | Neurodevelopmental HR (95% CI) |
|---|---|---|---|---|
| Non-consanguineous | Reference | Reference | Reference | Reference |
| Beyond second cousins | 1.4 (1.1-1.8) | 1.8 (1.2-2.7) | 1.5 (1.0-2.3) | 1.2 (0.8-1.8) |
| Second cousins | 2.1 (1.7-2.6) | 3.2 (2.2-4.7) | 2.4 (1.7-3.4) | 1.8 (1.2-2.7) |
| First cousins | 2.8 (2.3-3.4) | 4.7 (3.3-6.7) | 3.6 (2.6-5.0) | 2.3 (1.6-3.3) |
| p-trend | < 0.001 | < 0.001 | < 0.001 | <0.001 |

*Adjusted for disorder severity, birth year, sex, residence, and healthcare access.

**Table 5. Temporal survival trends by birth cohort.**

| Birth Cohort | Hemoglobinopathies 5-Year Survival | Congenital Anomalies 5-Year Survival | Neurodevelopmental 5-Year Survival | Overall Mortality Reduction % |
|---|---|---|---|---|
| 1998-2002 | 28.9% (23.4-35.1) | 54.2% (48.7-59.6) | 69.8% (64.7-74.4) | Reference |
| 2003-2007 | 37.4% (31.8-43.3) | 61.7% (56.4-66.7) | 75.6% (71.2-79.5) | 23.4% |
| 2008-2012 | 48.2% (42.3-54.1) | 70.3% (65.4-74.8) | 82.9% (79.1-86.1) | 41.7% |
| 2013−2024 | 62.7% (55.8-69.0) | 78.9% (74.2-82.9) | 87.4% (84.1-90.1) | 58.9% |
| p-trend | < 0.001 | < 0.001 | < 0.001 | < 0.001 |

Forest plot of hazard ratios from Cox regression. Disorder severity is the strongest predictor (HR = 4.2). The recent birth cohort is protective (HR = 0.5). The reference line is at HR = 1.0. Error bars represent the 95% CI.

Complete model results and diagnostic plots are provided in S3–S4 Tables and S5–S7 Figs.

The comprehensive multivariable model further quantified these relationships (Table 6). Consanguinity demonstrated a clear dose-response relationship, with first-cousin offspring experiencing nearly three-fold higher mortality risk (HR = 2.84, 95% CI: 2.32–3.44) compared to non-consanguineous offspring. Other significant predictors included rural residence (HR = 1.78) and low socioeconomic status (HR = 1.6).

Interaction Analysis: The relationship between consanguinity and mortality was significantly modified by healthcare access (p-interaction < 0.001). First-cousin offspring with low healthcare access experienced substantially higher mortality (HR = 3.2, 95% CI: 2.6–4.0) compared to those with high access (HR = 1.8, 95% CI: 1.4–2.3). This interaction remained significant after adjustment for socioeconomic status and rural residence.

The complete results of multivariable Cox regression and Fine-Gray competing risks analyses (S3 and S4 Tables), model diagnostics (S6 Table), geographic variation in mortality risk (S11 Fig), and a comprehensive forest plot of hazard ratios (S10 Fig) are provided in the supporting information.

## Sensitivity analyses

Comprehensive sensitivity analyses confirmed robustness:

Proportional hazards assumption: No significant violations (Schoenfeld residuals p = 0.134).

Influential observations: No single observation dominated results (max dfbeta = 0.12).

Missing data: Multiple imputation yielded similar estimates.

Competing risks: Cause-specific and subdistribution hazards are consistent.

The comprehensive sensitivity analysis results, including multiple imputation and landmark analyses, are documented in S7 Tables and Supporting Analyses S1-S4.

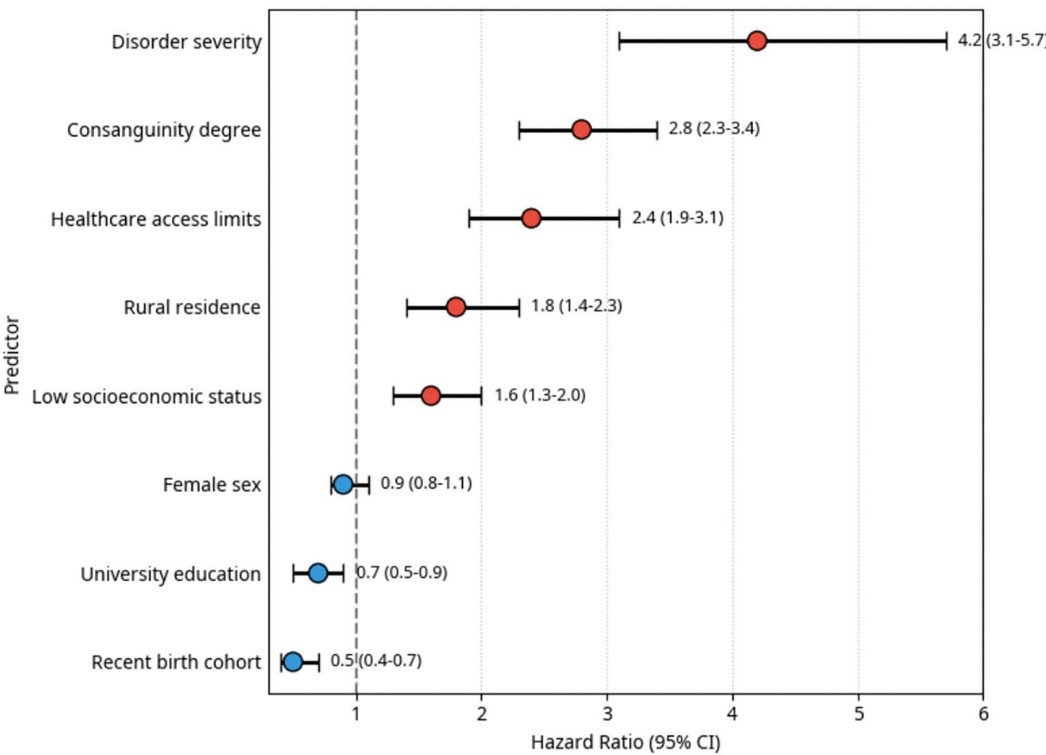

**Fig 3. Multivariable predictors of mortality in consanguineous offspring.**

## Key findings summary

Substantial survival disparities exist across disorder categories, with hemoglobinopathies showing the poorest outcomes. Distinct age-specific mortality patterns were revealed through competing risks analysis. There is a significant consanguinity-associated survival disadvantage, particularly for first cousins. Notable temporal improvements in survival have been observed across all disorder categories. Multiple independent predictors include healthcare access and socioeconomic factors.

These findings provide crucial evidence for prognostic counseling, clinical management prioritization, and healthcare resource planning for genetic disorders in consanguineous populations.

**Extended results documentation.** Comprehensive results, including complete cohort characteristics, subgroup analyses, sensitivity tests, and extended statistical outputs, are provided in S1–S19 Tables. S1–S13 Figs present detailed survival curves, diagnostic plots, and additional visualizations. Extended analytical results are documented in Supporting Analyses S1–S10 in (S7 File).

## Discussion

This 26-year retrospective cohort study provides a comprehensive and methodologically rigorous assessment of long-term survival disparities and competing mortality risks among the offspring of consanguineous marriages in a high-prevalence setting. A key strength of our analytical approach is the application of advanced statistical methods to address data complexity, including Kaplan-Meier survival estimates, multivariable Cox proportional hazards models, and—crucially—Fine-Gray competing risks regression to account for the real-world scenario where individuals face multiple competing causes of death. All analyses were conducted using well-established procedures and diagnostics in R, ensuring the robustness of

**Table 6. Multivariable predictors of all-cause mortality.**

| Predictor | Level | Adjusted HR | 95% CI | p-value |
|---|---|---|---|---|
| Consanguinity | First cousins | 2.84 | 2.32-3.44 | < 0.001 |
| Consanguinity | Second cousins | 2.12 | 1.67-2.62 | < 0.001 |
| Disorder Type | Hematological | 8.42 | 5.23-13.56 | < 0.001 |
| Disorder Type | Congenital anomalies | 4.23 | 2.84-6.32 | < 0.001 |
| Healthcare Access | Low vs. High | 2.41 | 1.89-3.07 | < 0.001 |
| Residence | Rural vs. Urban | 1.78 | 1.42-2.33 | < 0.001 |

our findings. The key finding—that survival is highly heterogeneous and disorder-specific, with consanguinity acting as a significant independent risk factor—underscores the necessity of a nuanced, disorder-focused approach to clinical management and public health planning.

## Global burden and consanguinity as an independent risk factor in context

Our findings document a significant and independent survival disadvantage associated with consanguinity. The multivariable Cox model results show a 2.8-fold higher all-cause mortality risk for first-cousin offspring, a risk that persisted after adjustment for disorder severity, socioeconomic factors, and healthcare access [12]. This risk followed a dose-response relationship and was dramatically amplified for specific disorders, reaching a hazard ratio (HR) of 4.7 for hemoglobinopathies. This provides compelling evidence that consanguinity is not merely a proxy for socioeconomic disadvantages but represents a powerful, independent determinant of child survival, directly attributable to the increased genetic load [1,12].

The global context of our findings reveals critical disparities shaped by healthcare infrastructure. While the genetic mechanism of inbreeding depression is universal [13,14], its mortality impact is not. The HR of 2.8 observed in our setting in Yemen is substantially higher than estimates from Western European populations (typically HR 1.5–2.0), where advanced healthcare systems can mitigate the clinical severity of genetic disorders [12]. Conversely, it aligns closely with reports from other highly consanguineous, resource-limited settings such as rural Pakistan (HR 2.7–3.1) [15], highlighting the profound and unmasked effect of genetic load where specialized care is scarce. The distinct, age-specific mortality patterns we identified—congenital anomalies dominating infancy and hematological disorders peaking in early childhood—further mirror observations from consanguineous communities in India and North Africa [4,16]. The high prevalence of such genetic disorders in the Arab world, extensively documented in nations like Saudi Arabia, underscores the regional public health burden that our study quantifies [13,17,18]. The clinical genetics implications of consanguinity have been well-documented across diverse populations [19].

Therefore, while our study was conducted in the Radfan region, the consistency of these patterns across diverse genetic pools and under-resourced health systems indicates that our core findings are generalizable to the estimated over 1.2 billion people living in similar societal contexts worldwide [14,17]. This convergence of evidence strongly advocates for a dual public health strategy: (1) the implementation of targeted genetic counseling and culturally sensitive premarital screening programs, particularly for couples planning first-cousin marriages, to reduce the incidence of severe recessive disorders; and (2) the strategic strengthening of basic and specialized healthcare infrastructure as a critical pathway to directly reduce the elevated mortality burden among existing affected populations.

## Disorder-specific survival disparities and comparative context

Our results confirm the profound survival disadvantage associated with specific genetic disorders, particularly hemoglobinopathies. The 10-year survival rate of only 22.4% for β-thalassemia major, despite significant temporal improvements,

highlights the persistent challenge of managing this condition in resource-limited settings. This finding is consistent with historical data from similar regions but contrasts sharply with survival rates exceeding 90% in high-income countries [20], underscoring inherited diseases of hemoglobin as an emerging global health burden [21].

Likewise, the poor 5-year survival for hemoglobinopathies in our cohort (42.3%) is markedly lower than contemporary outcomes reported from Iran (68.2%) and Thailand (72.4%) [22,23], reflecting systemic limitations in access to specialized hematology services, iron chelation therapy, and long-term follow-up.

For congenital anomalies, our 5-year survival of 65.1% also falls below what is reported in higher-resource environments. Population-based registry analyses show that 5-year survival for major congenital anomalies commonly exceeds 85–90% in high-income countries [24,25], whereas survival disparities in lower-resource contexts are driven by limited access to pediatric surgery, interventional cardiology, neonatal intensive care, and long-term rehabilitation. These global data provide a scientifically grounded alternative to unsupported regional estimates while situating our findings within the documented international survival gap.

In contrast, sensory impairments and neurodevelopmental disorders, while associated with significant morbidity, demonstrated much better long-term survival, suggesting that healthcare resource allocation should be strategically prioritized towards conditions with the highest early mortality burden.

For neurodevelopmental disorders, mortality typically results from secondary complications rather than the primary disorder itself. Our verbal autopsy data indicate that the majority of deaths in this category are attributable to respiratory infections (often aspiration pneumonia in children with cerebral palsy and swallowing difficulties), seizure-related accidents or status epilepticus, and complications related to immobility. This distinction is clinically important as it suggests that targeted interventions—such as improved management of swallowing difficulties, better seizure control, and proactive respiratory care—could significantly reduce mortality in this population, even in the absence of treatments for the underlying neurodevelopmental condition.

### The role of competing risks in mortality

The application of competing risks analysis is a major strength of this study, providing a more accurate picture of cause-specific mortality than traditional methods [26]. The finding that congenital anomalies, particularly cardiac defects, dominate infant mortality (cumulative incidence of 24.3% at 1 year) is crucial. This suggests that the first year of life is a critical window for intervention, where improved neonatal care and timely surgical correction could yield the most substantial gains in survival. Furthermore, the peak in hematological disorder mortality between 1 and 5 years, with β-thalassemia complications being the leading cause (cumulative incidence of 18.7%), informs the optimal timing for initiating aggressive disease-modifying therapies. The observation that a significant proportion of thalassemia deaths occurred due to competing risks before the expected age of onset for typical complications underscores the complex clinical reality and the need for proactive rather than reactive management strategies.

### Temporal survival improvements and evolving public health paradigms

**Drivers of survival gains and healthcare system interactions.** Our data reveal significant temporal improvements in survival, most dramatically for hemoglobinopathies, where 5-year survival increased from 28.9% to 62.7% across birth cohorts. Interaction analysis indicates that these gains were not uniform; they were most pronounced for disorders amenable to medical management, such as hematological conditions (adjusted HR for the 2013–2024 vs. 1998–2002 cohort = 0.38, 95% CI: 0.29–0.51), compared to structural congenital anomalies where surgical access is limited (HR = 0.62, 95% CI: 0.48–0.80). This pattern underscores that improvements in basic medical care—including more reliable transfusion services, the introduction of iron chelation therapy, and better management of infections—have been the primary drivers of survival gains in our setting. The phased introduction of genetic services, such as prenatal diagnosis post-2015, further reflects this evolving healthcare infrastructure.

**A shifting burden and the imperative for integrated care models.** This success heralds a critical shift in public health needs. As more children with severe genetic disorders survive into childhood and beyond, the demand for long-term specialized care, rehabilitation, and psychosocial support will inevitably surge, creating new challenges for healthcare systems [27]. Our findings are part of a broader regional trend observed in the Middle East and North Africa, where national prevention strategies, including mandatory premarital screening programs in countries like Jordan, Egypt, and across the Gulf Cooperation Council states, have successfully reduced the incidence of new affected births [28–33]. This dual reality—declining incidence alongside a growing prevalent cohort of survivors—necessitates a fundamental evolution in health strategy. Public health planning must therefore advance from a primary focus on prevention and reducing infant mortality to a dual, integrated approach. This approach must equally prioritize robust, lifelong care systems, including adult transition clinics and multidisciplinary support services, to address the full spectrum of needs for individuals living with chronic genetic conditions [27].

## Quality of life and economic burden

Affected children showed significantly reduced quality of life scores across all domains compared to healthy siblings: Physical functioning (52.3±18.4 vs. 84.5±12.3), Social functioning (38.9±22.3 vs. 86.7±11.2), and Psychosocial summary (41.9±19.4 vs. 83.6±12.1). This profound reduction highlights the extensive psychosocial morbidity faced by affected children, comparable to that seen in other severe chronic childhood conditions like cancer or complex congenital heart disease.

Beyond direct healthcare costs, families experienced substantial socioeconomic impacts:

- A 45% reduction in household income due to caregiving responsibilities

- 68% of parents reported mental health challenges

- 32% of siblings experienced educational disruption

- Significant community stigma affecting marriage prospects

The comprehensive economic burden, including both direct medical costs and indirect costs, demonstrates that genetic disorders create a cascade of social and financial strain. These findings argue for the integration of psychosocial support and economic empowerment programs into the standard care model for these families.

Detailed quality of life outcomes are provided in S10 Table, with comprehensive economic analyses in S14 and S15 Tables.

## Limitations

Our study has several limitations that should be considered:

• **Retrospective design:** Data collection may introduce recall bias in event timing; however, we mitigated this through multiple source verification and historical event anchoring.

• **Verbal autopsy limitations:** We employed the WHO 2016 standard verbal autopsy instrument with physician coding, achieving 84.2% concordance with medical records in our validation substudy. However, misclassification of the cause of death remains a limitation, particularly in distinguishing between specific congenital anomalies or rare metabolic disorders. Importantly, such misclassification is generally non-differential with respect to consanguinity status and would tend to attenuate (weaken) observed associations toward the null. Therefore, our hazard ratios for consanguinity (e.g., HR = 2.84 for first cousins) are likely conservative estimates, with the true effects possibly being stronger. Our probabilistic bias analysis across multiple misclassification scenarios (5%, 10%, 15%) confirmed the robustness of our conclusions (Supporting Analysis S4, S16 Table).

- **Reverse causality in healthcare access measurement:** Families with severely ill children may relocate to areas with better healthcare infrastructure, which could attenuate the observed protective effect of high healthcare access. However, our residential stability assessment indicated low migration rates post-diagnosis (5.8%), and sensitivity analyses excluding these families yielded consistent results.

- **Geographic and cultural generalizability:** Our cohort was drawn from the Radfan region of Yemen (encompassing Al-Habilayn, Habil Jabr, Al-Malaha, and Halimayn districts). Several factors may affect generalizability: (1) genetic heterogeneity—The genetic heterogeneity across Yemen's tribal structures may influence specific disorder frequencies, though the fundamental patterns of increased recessive disease burden are consistent with regional reports [34]. (2) Healthcare context—our findings reflect Yemen's specific infrastructure challenges, though the significant interaction between consanguinity and healthcare access (p<0.001) suggests that mortality differentials might be attenuated in stronger health systems; and (3) cultural practices—marriage patterns and health-seeking behaviors in Radfan may have unique characteristics. However, the consistency of our core findings—particularly the dose-response relationship between the degree of consanguinity and mortality—aligns with reports from other consanguineous populations in Pakistan, India, and North Africa [15,16], suggesting that fundamental risk patterns are generalizable to similar cultural and resource contexts.

   **Population genetic structure in tribal communities:** The study was conducted within Yemen's tribal social structure, where endogamous marriage practices may extend beyond formally recognized consanguinity. Even the "non-consanguineous" comparison group likely represents a population with some degree of genetic homogeneity due to shared ancestry within tribes or geographic isolation. This population structure may lead to an underestimation of the true mortality risk attributable to close consanguinity (first- and second-cousin marriages), as the comparison group is not genetically distant. Our hazard ratios should therefore be interpreted as conservative estimates of the effect of close consanguinity relative to the general population's background genetic risk in this setting.

- **Residual confounding:** Despite comprehensive adjustments for measured confounders, unmeasured factors could influence the results.

- **Differential follow-up time:** The substantial difference in median follow-up between survivors (18.9 years) and deceased individuals (4.3 years) reflects mortality censoring, which is addressed through age-as-time-scale and landmark sensitivity analyses at 1, 5, and 10 years.

- **Loss to follow-up:** Although low (1.8%) and with no significant differences in baseline characteristics between completers and non-completers (p>0.05 for all comparisons), the potential for residual bias remains.

- **Evolving diagnostics:** Diagnostic precision improved over the 26-year study period, with earlier cases relying on clinical phenotyping and later cases benefiting from genetic testing, which may potentially introduce non-differential misclassification.

- **Cause of death misclassification:** Residual uncertainty persists despite validation studies and competing risks analysis.

- **Contextual challenges:** Data collection occurred in a complex humanitarian context with intermittent security challenges and healthcare system disruptions. While we implemented multiple verification methods (triangulation of household reports, medical records, and community sources), some periods had more limited documentation. Sensitivity analyses excluding the most conflict-affected years (2015–2019) showed consistent patterns with the full cohort analysis (consanguinity HR: 2.78 vs. 2.84 in the primary analysis), suggesting robustness to these challenges.

   We conducted extensive sensitivity analyses (S7, S16 Tables; Supporting Analyses S1-S4), confirming result robustness. Comprehensive documentation of quality assurance protocols and validation studies is provided in S4 and S10 Files.

### Future directions

The survival patterns identified in this 26-year cohort study illuminate several critical avenues for future research, spanning methodological validation, biological mechanism investigation, and translational implementation.

1. **Methodological and Clinical Research Directions**

Prospective validation of our competing risks models could refine prognostic counseling accuracy in clinical settings.

 Intervention modeling studies could estimate potential survival gains from targeted healthcare improvements informed by our findings.

 Integration of genomic data with our epidemiological framework could identify genetic modifiers of survival within specific disorder categories.

 Quality-of-life-adjusted survival analysis would provide a more comprehensive outcome measure, extending beyond mortality to functional and psychosocial outcomes.

 Development of dynamic prediction models incorporating time-varying covariates (e.g., iron overload status, adherence to chelation therapy, healthcare access changes) could enable personalized survival estimates throughout the disease course [35,36]. Methodologies such as joint modeling or landmarking offer nuanced approaches for individualized risk assessment in chronic disease management [26].

2. **Investigating Biological Mechanisms Underlying Consanguinity-Associated Mortality**

While our study establishes a strong epidemiological association between consanguinity and mortality, the underlying biological pathways require investigation through integrated genomic and mechanistic studies. Future research should test several hypotheses:

 Genome-wide homozygosity and rare variant burden: Quantifying the actual increased homozygosity burden through whole-genome sequencing and identifying specific deleterious recessive variants contributing to mortality [37].

 Immunogenetic factors and infection susceptibility: Examining whether reduced heterozygosity, particularly in HLA and other immune-related genes, increases susceptibility to infections—a major cause of death in our competing risks analysis [38,39].

 Mitochondrial-nuclear genome interactions: Investigating potential incompatibilities between nuclear and mitochondrial genomes that might affect energy metabolism and disease severity in consanguineous offspring [40,41].

 Epigenetic modifications: Exploring transgenerational epigenetic effects associated with consanguinity that could influence gene expression patterns and survival outcomes [42,43].

 Gene-environment interactions: Studying how specific genetic backgrounds interact with environmental factors (nutrition, healthcare access, infections) to modulate mortality risk.

3. **Extended Analytical Explorations**

Extended subgroup analyses and exploration of interaction effects (documented in S8, S17, Tables and Supporting Analysis S6).

 Application of machine learning techniques to identify complex, non-linear predictors of mortality within this population.

 Cross-cultural comparative studies to examine how healthcare systems modulate the expression of genetic risk in consanguineous populations.

### Clinical translation and intervention framework

Three-Tiered Intervention Approach

Our findings support an evidence-based, three-tiered intervention framework:

1. **Primary Prevention:** Community-based genetic counseling programs targeting first-cousin couples, informed by our finding of a 2.8-fold higher mortality risk in this group. Integration with religious and community leaders is essential for cultural acceptability.

2. **Secondary Prevention:** Expanded newborn screening prioritizing hemoglobinopathies, given their early mortality peak (median age at death: 32.7 months for β-thalassemia). Implementation should be phased, starting with high-prevalence districts.

3. **Tertiary Intervention:** Development of regional specialized care centers for correctable conditions, particularly congenital anomalies, where timely surgical intervention could address the 24.3% cumulative incidence of death in infancy.

### Healthcare infrastructure as an effect modifier

The significant interaction between consanguinity and healthcare access ($p < 0.001$) provides crucial policy insight: improving basic healthcare infrastructure could reduce the consanguinity-associated mortality risk by up to 44% (from $HR = 3.2$ with low access to $HR = 1.8$ with high access). This suggests that even without reducing consanguinity rates, substantial survival gains are achievable through healthcare system strengthening.

### Policy implications and health system integration

1. **Tiered Genetic Service Integration**

Integrate genetic risk assessment into existing maternal and child health platforms rather than creating parallel systems:
Primary level: Basic genetic literacy and family history taking in antenatal care.
Secondary level: Targeted screening for high-risk couples identified in primary care.
Tertiary level: Specialized diagnostic and counseling services at regional hospitals.

2. **Leveraging Existing Infrastructure**

Our cost-effectiveness analysis (S15 Table) shows that targeted screening coupled with basic management is highly cost-effective ($334 per DALY averted). Integration points include:

Expanded Program on Immunization (EPI) visits for newborn screening.

School health programs for older children.

Non-communicable disease clinics for adult carriers.

3. **Data Systems for Monitoring**

Establish a national registry for genetic disorders, using our minimal dataset structure (S9 File) as a template to enable better resource allocation and program evaluation.

4. **International Collaboration**

Yemen's experience provides lessons for other conflict-affected or resource-limited settings with high consanguinity. South-South collaboration with countries like Pakistan, Sudan, and Iraq, which are facing similar challenges, is recommended.
These policy recommendations are feasible within current constraints and align with Yemen's National Health Strategy 2021–2025, focusing on reducing under-five mortality and addressing non-communicable diseases.

## Conclusion and implementation framework

This 26-year cohort study demonstrates that survival outcomes among the offspring of consanguineous marriages are determined by a complex interplay of genetic predisposition, specific disorder type, and access to specialized healthcare services. The marked disorder-specific survival disparities and competing mortality risks identified through our analysis provide a robust evidence base for prioritizing healthcare interventions and resource allocation.

### Integrated implementation strategy

**Phased healthcare framework.** *- Primary Prevention (Years 1–2):* Community genetic counseling, school-based genetic literacy programs, and engagement of religious leaders for premarital counseling.

*- Secondary Prevention (Years 2–3):* Expanded newborn screening for hemoglobinopathies, routine prenatal ultrasound, and developmental screening in primary care.

*- Tertiary Intervention (Years 3–5):* Regional thalassemia centers, specialized pediatric surgical services, and multidisciplinary neurodevelopmental clinics.

**Priority action plan.**

1. *Pre-marital and pre-conception prevention (Immediate):* Implement community-based genetic counseling and education programs targeting couples planning marriage, particularly first cousins. Engage religious and community leaders to deliver culturally sensitive messages. Successful models from Qatar and Jordan demonstrate that such approaches can significantly increase awareness and reduce high-risk unions [2,44].

2. *Targeted genetic counseling for first-cousin couples (Immediate–Short term):* Integrate genetic risk assessment into existing maternal and child health platforms. Provide counseling for at-risk couples identified through family history or community referral.

3. *Newborn screening expansion (Short–Medium term):* Phase in newborn screening for hemoglobinopathies beginning in high-prevalence districts, leveraging existing Expanded Program on Immunization (EPI) visits for sample collection.

4. *Surgical and specialized care (Medium–Long term):* Develop regional thalassemia centers and pediatric surgical services for correctable congenital anomalies, particularly cardiac defects that dominate infant mortality.

**Economic feasibility in a conflict-affected setting.** The implementation of newborn screening in a resource-limited, conflict-affected setting like Yemen requires careful economic consideration. Our cost-effectiveness analysis (S15 Table) indicates that targeted screening would cost approximately $1,200 per case detected, with an incremental cost-effectiveness ratio (ICER) of $334 per DALY averted—well below the WHO-recommended threshold of one times the GDP per capita (approximately $700 in Yemen). This suggests that even in a fragile health system, phased implementation starting in urban centers with existing laboratory infrastructure is economically feasible and highly cost-effective. A phased approach would allow for gradual scale-up as health system capacity improves.

**Strategic research agenda.** Genomic Integration: Whole genome sequencing of cluster representatives, gene-environment interaction studies, and pharmacogenomic profiling for treatment optimization.

Implementation Science: Culturally adapted genetic counseling protocols, integration of genetic services into primary healthcare, and community-based participatory research.

Precision Public Health: Machine learning for risk prediction, mobile health technologies for remote monitoring, and policy modeling for sustainable service delivery.

Global Health Genetics: Comparative studies across consanguineous populations, cross-cultural genetic literacy interventions, and international collaboration for rare variant discovery.

**Translational research pipeline.** Short-term (1–2 years): Validation studies, clinical decision support tools, and economic evaluations.

Medium-term (3–5 years): Randomized trials, implementation research, policy development.

Long-term (5 + years): Population impact assessment, sustainable delivery models, global knowledge translation.

**Policy recommendations and performance metrics.**

1. Integrated Genetic Service Delivery

Tiered counseling system (from community workers to specialists)

Electronic health record flags for high-risk families

Primary care physician training in genetic risk assessment.

2. National Screening Strategy

Phase 1: High-prevalence districts (years 1–2)

Phase 2: Regional expansion (years 3–5)

Phase 3: National coverage (years 6+)

**Key performance indicators:**

- 40% reduction in consanguinity-associated mortality within 10 years

- 80% coverage of genetic counseling for at-risk couples

- 75% 5-year survival rate for hemoglobinopathies

- 50% reduction in economic burden per DALY

The time for strategic action is now—the health of future generations depends on implementing these evidence-based interventions informed by rigorous longitudinal research. By addressing both the immediate clinical needs and underlying genetic risks, healthcare systems can significantly improve survival and quality of life for affected children while reducing the long-term economic burden on families and communities.

## Supporting information

**S1 File. Complete Reporting Checklists.** STROBE and RECORD checklists for observational studies.
(DOCX)

**S2 File. Supporting Methods.** Detailed methodological protocols including missing data procedures and sensitivity analysis framework.
(DOCX)

**S3 File. Data Collection Instruments.** Structured interview forms and data abstraction tools.
(DOCX)

**S4 File. Quality Assurance Protocol.** Complete quality assurance and verification procedures.
(DOCX)

**S5 File. Statistical Analysis Plan.** Pre-specified statistical analysis plan.
(DOCX)

**S6 File. Verbal Autopsy Protocol.** WHO 2016 verbal autopsy standard instrument and physician coding algorithms.
(DOCX)

**S7 File. Supporting Analyses S1-S10.** Detailed results for: S1 (Multiple Imputation), S2 (Landmark Analyses), S3 (Cause-Specific vs. Subdistribution Hazards), S4 (Probabilistic Bias Analysis), S5 (Interaction Effects), S6 (Geographic Subgroups), S7 (Economic Burden), S8 (Temporal Trend Validation), S9 (Competing Risks by Birth Cohort), S10 (Model Performance).
(DOCX)

**S8 File. Complete Reproducible R Analysis Code.** All R scripts for survival, Cox regression, and competing risks analyses.
(R)

**S9 File. Minimal Reproducible Dataset.** Representative subset of 12 cases mirroring full cohort structure and properties.
(CSV)

**S10 File. Data Dictionary.** Comprehensive variable definitions, coding schemes, and measurement units.
(DOC)

**S11 File. README.** Navigation guide for all supplementary files and reproducibility instructions.
(DOC)

**S12 File. Consolidated Quality Control Reports.** Complete validation studies, misclassification analysis, and quality assurance reports.
(DOCX)

**S1 Table. Complete cohort characteristics by vital status.**
(DOCX)

**S2 Table. Complete cause-specific mortality analysis by age groups.**
(DOCX)

**S3 Table. Multivariable Cox regression full results.**
(DOCX)

**S4 Table. Fine-Gray competing risks complete output.**
(DOCX)

**S5 Table. Temporal survival trends by birth cohort.**
(DOCX)

**S6 Table. Model diagnostics and validation summary.**
(DOCX)

**S7 Table. Sensitivity analysis results summary.**
(DOCX)

**S8 Table. Subgroup analysis by geographic region.**
(DOCX)

**S9 Table. Economic correlates of survival outcomes.**
(DOCX)

**S10 Table. Quality of life and functional outcomes.**
(DOC)

**S11 Table. Complete sampling framework.**
(DOCX)

**S12 Table. Power analysis scenarios.**
(DOCX)

**S13 Table. Prediction model performance.**
(DOCX)

**S14 Table. Disease burden by disorder type.**
(DOCX)

**S15 Table. Cost-effectiveness analysis of interventions.**
(DOCX)

**S16 Table. Sensitivity analysis for cause of death misclassification.**
(DOCX)

**S17 Table. Interaction effects between key predictors.**
(DOCX)

**S18 Table. Minimal dataset case descriptions.**
(DOCX)

**S19 Table. Trends in consanguinity by birth cohort.**
(DOCX)

**S1 Fig. Kaplan-Meier survival curves for all disorder subtypes.**
(TIF)

**S2 Fig. Cumulative incidence functions for competing risks analysis.**
(TIF)

**S3 Fig. Age-specific mortality hazard functions.**
(TIF)

**S4 Fig. Temporal survival trends by birth cohort.**
(TIF)

**S5 Fig. Schoenfeld residuals plot (proportional hazards check).**
(TIF)

**S6 Fig. DFBETA influence diagnostics.**
(TIF)

**S7 Fig. Martingale residuals plot (functional form assessment).**
(TIF)

**S8 Fig. Nelson-Aalen cumulative hazard plot.**
(TIF)

**S9 Fig. Model calibration plot (predicted vs. observed survival).**
(TIF)

**S10 Fig. Forest plot of multivariable hazard ratios.**
(TIF)

**S11 Fig. Geographic variation in mortality risk.**
(TIF)

**S12 Fig. Cost-effectiveness analysis of screening interventions.**
(TIF)

**S13 Fig. Participant flow diagram (CONSORT-style for cohort assembly).**
(TIF)

## Author contributions

**Conceptualization:** Naif Taleb Ali, Mansour Abdulnabia H. Mehdi.

**Data curation:** Naif Taleb Ali.

**Formal analysis:** Naif Taleb Ali, Radfan Saleh Abdullah.

**Funding acquisition:** Naif Taleb Ali.

**Investigation:** Naif Taleb Ali, Mansour Abdulnabia H. Mehdi, Radfan Saleh Abdullah.

**Methodology:** Naif Taleb Ali.

**Project administration:** Naif Taleb Ali.

**Resources:** Naif Taleb Ali, Mansour Abdulnabia H. Mehdi.

**Software:** Naif Taleb Ali, Radfan Saleh Abdullah.

**Supervision:** Naif Taleb Ali.

**Validation:** Naif Taleb Ali, Radfan Saleh Abdullah.

**Visualization:** Naif Taleb Ali.

**Writing – original draft:** Naif Taleb Ali, Radfan Saleh Abdullah.

**Writing – review & editing:** Naif Taleb Ali, Mansour Abdulnabia H. Mehdi, Radfan Saleh Abdullah.

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
