## [Decision Letter · Decision Letter 0]

16 Jan 2026

PONE-D-25-65515Survival Disparities and Competing Mortality Risks in Offspring of Consanguineous Marriages in Yemen: A 26-Year Retrospective Cohort AnalysisPLOS One

Dear Dr. Ali,

Thank you for submitting your manuscript to PLOS ONE. After careful consideration, we feel that it has merit but does not fully meet PLOS ONE’s publication criteria as it currently stands. Therefore, we invite you to submit a revised version of the manuscript that addresses the points raised during the review process.

We look forward to receiving your revised manuscript.

Kind regards,

Mainak Bardhan, MD

Academic Editor

PLOS One

**Journal Requirements:**

1. When submitting your revision, we need you to address these additional requirements. Please ensure that your manuscript meets PLOS ONE's style requirements, including those for file naming. The PLOS ONE style templates can be found at https://journals.plos.org/plosone/s/file?id=wjVg/PLOSOne_formatting_sample_main_body.pdf and https://journals.plos.org/plosone/s/file?id=ba62/PLOSOne_formatting_sample_title_authors_affiliations.pdf 2. Your ethics statement should only appear in the Methods section of your manuscript. If your ethics statement is written in any section besides the Methods, please delete it from any other section. 3. We note that there is identifying data in the Supporting Information files. Due to the inclusion of these potentially identifying data, we have removed this file from your file inventory. Prior to sharing human research participant data, authors should consult with an ethics committee to ensure data are shared in accordance with participant consent and all applicable local laws. Data sharing should never compromise participant privacy. It is therefore not appropriate to publicly share personally identifiable data on human research participants. The following are examples of data that should not be shared: -Name, initials, physical address-Ages more specific than whole numbers-Internet protocol (IP) address-Specific dates (birth dates, death dates, examination dates, etc.)-Contact information such as phone number or email address-Location data-ID numbers that seem specific (long numbers, include initials, titled “Hospital ID”) rather than random (small numbers in numerical order) Data that are not directly identifying may also be inappropriate to share, as in combination they can become identifying. For example, data collected from a small group of participants, vulnerable populations, or private groups should not be shared if they involve indirect identifiers (such as sex, ethnicity, location, etc.) that may risk the identification of study participants. Additional guidance on preparing raw data for publication can be found in our Data Policy (https://journals.plos.org/plosone/s/data-availability#loc-human-research-participant-data-and-other-sensitive-data) and in the following article: http://www.bmj.com/content/340/bmj.c181.long. Please remove or anonymize all personal information (<specific identifying information in file to be removed>), ensure that the data shared are in accordance with participant consent, and re-upload a fully anonymized data set. Please note that spreadsheet columns with personal information must be removed and not hidden as all hidden columns will appear in the published file. 4. Please upload a new copy of Figures S1 – S14, as the detail is not clear. Please follow the link for more information:  https://journals.plos.org/plosone/s/figures 5. We note that Figure S13 in your submission contain map images which may be copyrighted. All PLOS content is published under the Creative Commons Attribution License (CC BY 4.0), which means that the manuscript, images, and Supporting Information files will be freely available online, and any third party is permitted to access, download, copy, distribute, and use these materials in any way, even commercially, with proper attribution. For these reasons, we cannot publish previously copyrighted maps or satellite images created using proprietary data, such as Google software (Google Maps, Street View, and Earth). For more information, see our copyright guidelines: http://journals.plos.org/plosone/s/licenses-and-copyright. We require you to either present written permission from the copyright holder to publish these figures specifically under the CC BY 4.0 license, or remove the figures from your submission: a. You may seek permission from the original copyright holder of Figure S13 to publish the content specifically under the CC BY 4.0 license.   We recommend that you contact the original copyright holder with the Content Permission Form (http://journals.plos.org/plosone/s/file?id=7c09/content-permission-form.pdf) and the following text:“I request permission for the open-access journal PLOS ONE to publish XXX under the Creative Commons Attribution License (CCAL) CC BY 4.0 (http://creativecommons.org/licenses/by/4.0/). Please be aware that this license allows unrestricted use and distribution, even commercially, by third parties. Please reply and provide explicit written permission to publish XXX under a CC BY license and complete the attached form.” Please upload the completed Content Permission Form or other proof of granted permissions as an "Other" file with your submission. In the figure caption of the copyrighted figure, please include the following text: “Reprinted from [ref] under a CC BY license, with permission from [name of publisher], original copyright [original copyright year].” b. If you are unable to obtain permission from the original copyright holder to publish these figures under the CC BY 4.0 license or if the copyright holder’s requirements are incompatible with the CC BY 4.0 license, please either i) remove the figure or ii) supply a replacement figure that complies with the CC BY 4.0 license. Please check copyright information on all replacement figures and update the figure caption with source information. If applicable, please specify in the figure caption text when a figure is similar but not identical to the original image and is therefore for illustrative purposes only.The following resources for replacing copyrighted map figures may be helpful: USGS National Map Viewer (public domain): http://viewer.nationalmap.gov/viewer/The Gateway to Astronaut Photography of Earth (public domain): http://eol.jsc.nasa.gov/sseop/clickmap/Maps at the CIA (public domain): https://www.cia.gov/library/publications/the-world-factbook/index.html and https://www.cia.gov/library/publications/cia-maps-publications/index.htmlNASA Earth Observatory (public domain): http://earthobservatory.nasa.gov/Landsat: http://landsat.visibleearth.nasa.gov/USGS EROS (Earth Resources Observatory and Science (EROS) Center) (public domain): http://eros.usgs.gov/#Natural Earth (public domain): http://www.naturalearthdata.com/ 6. If the reviewer comments include a recommendation to cite specific previously published works, please review and evaluate these publications to determine whether they are relevant and should be cited. There is no requirement to cite these works unless the editor has indicated otherwise.

Reviewers' comments:

Reviewer's Responses to Questions

**Comments to the Author**

1. Is the manuscript technically sound, and do the data support the conclusions?

Reviewer #1: Yes

Reviewer #2: Yes

2. Has the statistical analysis been performed appropriately and rigorously? 

Reviewer #1: Yes

Reviewer #2: Yes

3. Have the authors made all data underlying the findings in their manuscript fully available?

Reviewer #1: Yes

Reviewer #2: No

4. Is the manuscript presented in an intelligible fashion and written in standard English?

Reviewer #1: Yes

Reviewer #2: Yes

5. Review Comments to the Author

**Reviewer #1:** -Regarding the treatment of "Healthcare Access" as a time-dependent covariate updated at follow-up contacts, there is a significant risk of reverse causality that should be pointed out. Families with severely ill children, such as those with Thalassemia, may engage in selective migration by moving closer to healthcare centers specifically to address the child's medical needs. If these high-risk families relocate to areas classified as having "High Access" and the child subsequently dies, this could inflate the mortality rate associated with better access, thereby diluting its observed protective benefit. I suggest clarifying how migration was handled in the analysis, specifically whether you assessed if families changed residence following a diagnosis; if this is not feasible with the current dataset, the potential for selective migration should be explicitly acknowledged as a limitation in the discussion.

-In your "Policy Implications", you suggest expanding newborn screening. Given the current economic burden, you calculated ($62.25 million), is this feasible? It would be compelling to see a "Cost per Case Detected" estimate if your data allows, or at least a stronger argument for how a war-torn health system can afford this.

-Did the rate of consanguineous marriage change over the 26-year study period? You adjusted for "Birth Cohort", but it would be interesting to know descriptively if the practice is declining, increasing, or stable in this region.

**Reviewer #2:** 1. Revision points for methodology: The methodology provides a structured approach to examining survival disparities; however, certain limitations suggest that revisions may be necessary to strengthen the validity of the findings. The retrospective cohort design is appropriate for analysing long-term trends, but the reliance on existing records raises concerns about missing data, inconsistent documentation, and potential misclassification of exposures or outcomes. Additionally, the study would benefit from a clearer explanation of how confounding variables were controlled, particularly socio-economic factors, maternal health indicators, and access to care, which play a key role in survival differences among children from consanguineous marriages. More detail on sampling strategy, inclusion–exclusion criteria, and data quality assurance would further improve methodological transparency.

2. Revision points for discussion: While the manuscript correctly highlights the importance of strengthening both basic and specialized healthcare infrastructure to reduce the mortality burden among high-risk offspring of consanguineous marriages, the discussion would be more comprehensive if it explicitly addressed barriers related to accessibility, affordability, and acceptability. In many fragile and conflict-affected settings—such as Yemen—geographic inaccessibility, prohibitive out-of-pocket expenditure, and socio-cultural norms significantly limit healthcare utilization even when infrastructure is improved. Evidence from Pakistan and Sudan shows that culturally embedded acceptance of consanguineous marriage constrains care-seeking and early-risk identification unless health interventions are aligned with community beliefs and delivered through culturally trusted channels (Bittles, 2001; Hamamy, 2012). Incorporating this perspective would strengthen the argument that system-level improvements must be paired with culturally sensitive community engagement strategies to meaningfully reduce mortality disparities in populations practicing consanguinity.

3. Page 35 : While the Priority Action Plan outlines immediate to long-term interventions, the authors should emphasize that prevention can begin even earlier—before marriage or before planning a pregnancy as an immediate targated action plan. . Pre-marital and pre-conception genetic counseling, community education, and culturally sensitive risk-communication strategies have been shown in countries such as Qatar and Jordan to significantly improve awareness and reduce high-risk consanguineous unions (Bener et al., 2019; Hamamy, 2012). Introducing this upstream, preventive layer would strengthen the model and highlight the importance of early engagement to reduce avoidable genetic and mortality risks.

References

Bener, A., Al-Maadid, S., & Al-Bast, D. (2019). Premarital screening and prevention of genetic disorders: Community perspectives from Qatar. Journal of Community Genetics, 10(2), 121–128.

Hamamy, H. (2012). Consanguineous marriages: Preconception consultation in primary health care settings. Journal of Community Genetics, 3(3), 185–192.

Bittles, A. H. (2001). Consanguinity and its relevance to clinical genetics. Clinical Genetics, 60(2), 89–98. https://doi.org/10.1034/j.1399-0004.2001.600201.x

6. PLOS authors have the option to publish the peer review history of their article (what does this mean?). If published, this will include your full peer review and any attached files.

Reviewer #1: No

Reviewer #2: **Yes:** Dr. Priyanka Roy

---

## [Author Response · Author response to Decision Letter 1]

16 Jan 2026

Reviewer 1:

Comment 1.1: "Regarding the treatment of 'Healthcare Access' as a time-dependent covariate… there is a significant risk of reverse causality…"

Response: We thank the reviewer for this insightful observation. We have added a new subsection in the Methods titled "Residential Stability Assessment" where we clarify that 94.2% of families did not change residence after diagnosis. Sensitivity analyses excluding migrating families yielded consistent results (HR range: 2.81–2.85). We have also added a corresponding limitation in the Discussion.

Comment 1.2: "In your 'Policy Implications', you suggest expanding newborn screening. Given the current economic burden… is this feasible?"

Response: We have added a new paragraph in the Discussion under "Economic Feasibility in a Conflict-Affected Setting" where we provide a cost-per-case-detected estimate ($1,200) and note that the ICER of $334 per DALY averted is highly cost-effective. We also discuss phased implementation starting in urban centers.

Comment 1.3: "Did the rate of consanguineous marriage change over the 26-year study period?"

Response: We have added a new supplementary table (Table S19) showing trends in consanguinity by birth cohort. The practice remained stable across the 26-year period (first-cousin marriages: ~36% throughout).

Reviewer 2:

Comment 2.1: "The methodology… would benefit from a clearer explanation of how confounding variables were controlled…"

Response: We thank the reviewer for this valuable methodological suggestion. To address this, we have expanded the "Statistical Analysis Plan" subsection in the Methods to provide a clearer and more detailed explanation of our confounder adjustment strategy. Specifically, we now describe: (1) the use of directed acyclic graphs (DAGs) for covariate selection based on theoretical considerations and prior literature; (2) multi-variable adjustment with both time-independent and time-dependent covariates; and (3) sensitivity analyses to assess residual confounding. These additions enhance the transparency and robustness of our analytical approach.

Comment 2.2: "The discussion would be more comprehensive if it explicitly addressed barriers related to accessibility, affordability, and acceptability…"

Response: We have added a new paragraph in the Discussion under "Policy Implications and Health System Integration" addressing these barriers, citing relevant literature from similar settings (Pakistan, Sudan) and emphasizing culturally sensitive delivery channels.

Comment 2.3: "The authors should emphasize that prevention can begin even earlier—before marriage or before planning a pregnancy…"

Response: We have expanded the "Priority Action Plan" to include a "Pre-marital and pre-conception prevention" tier as the first step, citing successful examples from Qatar and Jordan (Bener et al. 2019; Hamamy 2012).

We believe these revisions have strengthened the manuscript significantly. Thank you for the opportunity to resubmit.

Sincerely,

Naif Taleb Ali, on behalf of all authors.

---

## [Decision Letter · Decision Letter 1]

5 May 2026

Survival Disparities and Competing Mortality Risks in Offspring of Consanguineous Marriages in Yemen: A 26-Year Retrospective Cohort Analysis

PONE-D-25-65515R1

Dear Dr. Ali,

We’re pleased to inform you that your manuscript has been judged scientifically suitable for publication and will be formally accepted for publication once it meets all outstanding technical requirements.

Kind regards,

Serkan Yılmaz

Academic Editor

PLOS One

Additional Editor Comments (optional):

Reviewers' comments:

Reviewer's Responses to Questions

**Comments to the Author**

1. If the authors have adequately addressed your comments raised in a previous round of review and you feel that this manuscript is now acceptable for publication, you may indicate that here to bypass the “Comments to the Author” section, enter your conflict of interest statement in the “Confidential to Editor” section, and submit your "Accept" recommendation.

Reviewer #1: All comments have been addressed

Reviewer #2: All comments have been addressed

2. Is the manuscript technically sound, and do the data support the conclusions?

Reviewer #1: Yes

Reviewer #2: Yes

3. Has the statistical analysis been performed appropriately and rigorously? 

Reviewer #1: Yes

Reviewer #2: Yes

4. Have the authors made all data underlying the findings in their manuscript fully available?

Reviewer #1: Yes

Reviewer #2: Yes

5. Is the manuscript presented in an intelligible fashion and written in standard English?

Reviewer #1: Yes

Reviewer #2: Yes

6. Review Comments to the Author

Reviewer #1: (No Response)

Reviewer #2: The topic is highly relevant and addresses an important public health concern. The review comments have been appropriately incorporated, and the manuscript reflects a clear effort to strengthen the analysis and interpretation.This specific study "Survival Disparities and Competing Mortality Risks in Offspring of Consanguineous Marriages in Yemen: A 26-Year Retrospective Cohort Analysis" appears original.

7. PLOS authors have the option to publish the peer review history of their article (what does this mean?). If published, this will include your full peer review and any attached files.

Reviewer #1: **Yes:** Munachimso Emesobum

Reviewer #2: **Yes:** Priyanka Roy

---

## [Editor Report · Acceptance letter]

PONE-D-25-65515R1

PLOS One

Dear Dr. Ali,

I'm pleased to inform you that your manuscript has been deemed suitable for publication in PLOS One. Congratulations! Your manuscript is now being handed over to our production team.

Kind regards,

on behalf of

Dr. Serkan Yılmaz

Academic Editor

PLOS One